# Transposable Elements as a Source of Novel Repetitive DNA in the Eukaryote Genome

**DOI:** 10.3390/cells11213373

**Published:** 2022-10-26

**Authors:** Michelle Louise Zattera, Daniel Pacheco Bruschi

**Affiliations:** 1Departamento de Genética, Programa de Pós-Graduação em Genética, Setor de Ciências Biológicas, Universidade Federal do Paraná, Curitiba 81530-000, PR, Brazil; 2Departamento de Genética, Laboratorio de Citogenética Evolutiva e Conservação Animal, Setor de Ciências Biológicas, Universidade Federal do Paraná, Curitiba 81530-000, PR, Brazil

**Keywords:** satellite DNA, centromeric region, TE life cycle, tandem repeats

## Abstract

The impact of transposable elements (TEs) on the evolution of the eukaryote genome has been observed in a number of biological processes, such as the recruitment of the host’s gene expression network or the rearrangement of genome structure. However, TEs may also provide a substrate for the emergence of novel repetitive elements, which contribute to the generation of new genomic components during the course of the evolutionary process. In this review, we examine published descriptions of TEs that give rise to tandem sequences in an attempt to comprehend the relationship between TEs and the emergence of de novo satellite DNA families in eukaryotic organisms. We evaluated the intragenomic behavior of the TEs, the role of their molecular structure, and the chromosomal distribution of the paralogous copies that generate arrays of repeats as a substrate for the emergence of new repetitive elements in the genome. We highlight the involvement and importance of TEs in the eukaryote genome and its remodeling processes.

## 1. Introduction

The duplication and mobilization cycles of transposable elements (TEs) are evolutionary processes that enrich the paralogous copies that constitute the repetitive DNA content of the eukaryote genome. This repetitive DNA can be subdivided into two categories: tandem repeat sequences, which represent copies organized in juxtaposition to one another (e.g., DNA satellites, minisatellites, and microsatellites), or dispersed sequences, which include the transposable elements themselves [1,2].

In most cases, transposable elements and tandem repeats are studied independently, but evidence has been found that TEs may be involved in the origin of a library of tandem repeats that is typically dispersed throughout the eukaryote genome, where it plays a fundamental evolutionary role [3,4]. The sequences of homologies of satDNAs and transposons or retrotransposons that have been identified in many species indicate the existence of an intimate evolutionary relationship between the TEs and the emergence of tandem repeat sequences, which implies that the TEs are involved in the reshaping of the genomic architecture [3,5].

The conversion of one type of repetitive element into another has been reported in all the principal branches of the eukaryote tree of life [6,7,8]. Here, we review case studies of the emergence of repetitive DNA, focusing on the new satellite DNA (satDNA) families derived from TEs with the aim of understanding (i) the intragenomic behavior of the TEs, (ii) the role of their molecular structure, and (iii) the chromosomal distribution of the paralogous copies. We discuss the genetic mechanisms that produce the TE copies and contribute to the molecular co-option of these elements during the evolution of the genome and chromosomes, most of which are revealed by the interplay and turnover of the different repetitive classes in the genome.

There are two major categories of transposable elements: classes I and II. Class I elements, which are also known as retrotransposons, are dependent on the RNA for their transposition in the genome, while class II elements (DNA transposons) do not depend on any retrotranscription mechanism for their mobility [9].

The recruitment of copies of the TEs as a substrate for the evolutionary emergence of new tandem repeats and satDNA families is directly dependent on the behavior of these elements in the host genome. The intra-genomic behavior of the TEs in the host genome follows a four-phase life cycle (Figure 1), starting with (i) the birth or invasion phase, when the TEs are inserted into a new locus, followed by (ii) amplification, (iii) maturation, and finally, (iv) death or degeneration [10,11,12].

The birth or invasion phase (Figure 1—T1) occurs when a new chromosomal locus acquires at least one copy of a TE. In most cases, the TE is transmitted vertically through the evolutionary lineage by splitting from the ancestral species, while invasion occurs through new transposition events, either by the recombination of sequences already present in the genome or events of horizontal transfer [13,14].

As a new repeat unit of a class I element (retrotransposon) is established in the genome, the number of copies begins to increase rapidly (amplification phase; T3 in Figure 1). These paralogous copies, generated by retrotransposition, are distributed randomly throughout the genome [15,16]. Genomes of all characterized higher eukaryotes possess examples of transposable element (TE) bursts [17]. Burst events can cause a local accumulation of TEs in the chromosomes, generating extensive clusters of TEs [18,19,20,21].

Events of this type are more frequent in germ cells due to the temporary relaxation of the epigenetic control of the TEs during the early development of the germline. This opens a temporal window that may allow these elements to escape from their constraints and propagate in the host genome [22,23]. Amplification occurs primarily in male germ cells due to the continuous spermatogenesis occurring throughout the life span of the animal, whereas female germlines are arrested in meiosis and do not undergo the temporary relaxation of epigenetic control observed in the male germline [24,25].

By contrast, the repair mechanism of double-strand (subclass I) or single-strand breaks (subclass II; see [9]) that occur following mobilization events is a central genetic process that results in the addition of a number of copies of the DNA transposons, given that the chromosomal gap generated by the excision of a transposon sequence is repaired by a homologous recombination mechanism, which results in the reintroduction of the transposon at the donor site [26]. If the mobilization event occurs during the S phase, the homologous recombination can use a sister chromatid as a template for the repair of the chromosome gap, which also restores the excised element and results in an increase in the number of copies of the class II elements [26,27].

The maturation phase (T4 in Figure 1) is the most prolonged stage of the process, and is the most subject to the action of mechanisms of molecular evolution. The host genome can evolve defenses to stop the spread of the TEs by temporary epigenetic inactivation, irrespective of the mobilization mechanism involved in the insertion of the new TE [28]. During this phase, the silencing of transposition activity and the relaxation of selection pressures on the paralogous copies contribute to the accumulation of random mutations in the molecular structure of the TE sequences [29,30].

One of these pathways is through the silencing of the RNA by small RNA molecules—endogenous siRNAs (endo-siRNAs) or PIWI-associated RNAs (piRNAs)—which guide the process by modulating chromatin states or targeting the degradation of the RNA [31]. The piRNA pathway suppresses transposon activity in the metazoan germline, whereas the endo-siRNAs repress TE activity in the somatic tissue [31,32].

The transposition of the TEs may also be suppressed by DNA methylation, in which a methyl group is added covalently to the C’5 position of the cytosine (5-methylcytosine) [31,33]. This methylation is associated with repression of the TE, as first observed in the retrotransposon Activator (Ac), Suppressor-mutator (Spm), and Mutator (Mu) of maize [34,35,36]. This epigenetic suppression by DNA methylation is thought to contribute not only to the silencing of transcription, but also to the formation of the heterochromatic regions of the genome [37,38]. Epigenetic histone modifications, such as acetylation, phosphorylation, and methylation, may also be involved in the repression of the TEs [39]. In the genus *Arabidopsis*, for example, the histone methylation of H3K9me2 and H3K27me1 appears to contribute to the transcriptional silencing of some TEs [40], while histone deacetylase triggers the transcriptional activity of the different TEs [41]. The long-time prevalence of TEs under epigenetic control results in the progressive accumulation of mutations (substitutions or indels), which leads, in turn, to the degeneration of the sequence and a loss of identity, which either reduces or eliminates its capacity for amplification in the genome (Figure 1—T3). Recombination is often suppressed in the heterochromatic regions [42,43], which appears to be one of the reasons why these portions of the chromosome are prone to the accumulation of TEs, maintained by unequal crossing-over or genetic drift [44,45].

Stochastic mutations that interfere with the mobilization capacity of the silenced TE sequences are either fixed or lost primarily through genetic drift, which may represent the degeneration or death phase of the respective locus (Figure 1—T4). In this case, the heterochromatin is often the “final resting place” of dysfunctional TEs, due to the low frequency of recombination in these regions [44,45], leading to the formation of a “TE cemetery” [45,46], where the TEs may play an important role in the structure of the heterochromatin itself [47]. Given the high level of degeneration of the TE sequences, the recognition of genomes by searching for homologies often recovers contigs that are not very similar to the active copies. This hampers the understanding of their evolutionary history in the host genome.

The insertion and accumulation of some TEs in the vicinity of genes, for example, or their degenerate copies, represent an opportunity for molecular evolution or domestication [48,49,50]. Numerous studies have revealed protein motifs or repetitive regions of TEs being recruited for new genomic functions, providing an important source of novel sequences for the evolution of the genome [51,52,53]. New regulatory sequences and new protein-coding or even non-coding RNAs may play a beneficial role in the host genome [11,54,55], and may also be involved in a number of different processes of the regulation of gene expression. These processes include the regulation of transcription by the domesticated LTR retrotransposons and the action of the Amniota SINE1 element, which enhances the genes encoding the fibroblast growth factor of the inhibitor of an apoptosis protein family [56,57].

Given this, the ability of the TEs to provide a substrate for evolutionary innovations has attracted considerable interest over the years, and has been the focus of numerous studies of eukaryote species, which have found evidence of their involvement in the evolution of the genome through their reorganization.

## 2. Transposons as a Source of Repetitive Units for the Emergence of Tandem Repeats

A number of studies have provided examples of TE sequences that give rise to new repetitive classes, such as microsatellites, minisatellites, and satellite DNA [3,5,11,54,58]. Transposable elements represent an important substrate for genomic remodeling, and the emergence of new repetitive sequences, which generate a library of short repeat arrays that will subsequently be dispersed through the genome, to eventually become novel tandem repeats [59] (Figure 1—T4).

Transposable elements may contain sites predisposed for the formation of microsatellite DNA, which favors the dispersal of these repetitive units in the genome [5,58,60]. In the human genome, for example, approximately 23% of all tandem arrays (satellite, mini- and microsatellite sequences) are derived from TEs [61].

### 2.1. Classes and Mobility of the TEs

Studies of a range of different organisms have revealed a subtle prevalence of class I elements as a source of repetitions in the genome. Retrotransposons of the SINE (Short Interspersed Nuclear Element) superfamily, in particular the Alu elements, are a compelling example of this process. The non-autonomous SINEs, which range in size from 100 to 600 base pairs (bps), are widely distributed in the eukaryotes, where they play an important role in the organization of the genome, given their involvement in cell survival during many different types of physiological stress, for example [62]. The Alu elements, which are primate-specific, are considered to be the most widespread of the transposable elements, representing approximately 11% of the human genome [63,64,65]. These elements are considered to be the origin of the pλg3 satDNA and (GAA)n-type repeats, which are both found in the human genome [66,67,68], and the A-rich microsatellites in the primates [69]. Approximately 7276 minisatellites in the human genome have been derived from transposable elements, and 2663 are associated with Alu elements [61].

Non-autonomous TEs prevail as a source of the sequences for the formation of novel repetitive elements (Table 1), and the MITE (Miniature Transposable Elements with inverted repetitions) elements may be an especially interesting group for the understanding of this process. The MITE-type elements are non-autonomous, class II elements of approximately 400 bps, which are characterized by inverted repetitions flanking a variable region [9,70,71]. The MITE repeats are known to give rise to (GTCY)n repeats in lepidopterans [72], Xstir satDNA in *Xenopus leavis* [73], miDNA4 in *Xenopus tropicalis* [74], D1100 satDNA in rye [75], a number of different types of satellite DNA in *Messor* ants [76], and HindIII satDNA in bivalves [77].

The MITE sequences experience burst events, which may lead to a dramatic increase in the number of copies, resulting in the rapid accumulation of arranged units with highly similar sequences in the chromosomes [100,101,102], as seen in the amphibian *Xenopus tropicalis* [74]. In the genome of *X. tropicalis*, the presence of a miDNA4-MITE that contains a satellite DNA motif, demonstrates the synergic interplay between the MITE structure and genomic context, which contributes to the emergence of the repetitive arrays of the new satDNA. In this model, the birth of the satellite monomer within a MITE is followed by the amplification of tandem repeats, with the higher recombination rates of the tandem arrays of paralogous copies leading to the rapid homogenization of the repetitive arrays and, over time, the generation of satDNA following the putative concerted evolution model [74].

Helitron elements (subclass II of the DNA transposons) also represent a source of the spread of satDNA-like arrays in the genome. The transposition of these Class II elements occurs through semi-replicative transposition, in which only one strand of the transposon is transferred between genomic sites without duplicating the target site (TSD) upon insertion, a process known as Rolling Circle Transposition, or RCT [2,9,103,104]. Analysis of the RCT mechanism has revealed cases in which the rolling-circle transposases are unable to recognize the termination sites located at the extremity of each TE, which results in the transposition of fragments of the genomic DNA located in the immediate vicinity of the sequence. This makes the TEs prone to the capture and propagation of a range of different genomic sequences [2,105].

As satDNA may be formed by the tandem amplification of a whole TE, or only a part of it, where fractions of short satDNA-like arrays would be expected to be found dispersed throughout the genome as an intermediate stage of the emergence of satDNA from the TEs, which are normally distributed in euchromatic regions [2,8,74,106]. In *Crassostrea gigas*, for example, the genome assembly presents 13 clusters of satDNA-like tandem repeats, which represent the central repeats of 11 non-autonomous elements belonging to the Helentron superfamily of DNA transposons known as the CgHINE [2]. The genome-wide distribution of this element in this species indicates that Helentrons are able to propagate tandem repeats.

### 2.2. Are Certain Portions of a TE More Prone to the Generation of Tandem Repeat Sequences?

Although the entire sequence of a transposable element has the potential to act as the substrate for the generation of new repetitive elements [61], we observed a prevalence of the repetitive portions of these elements as the source of the monomeric units of tandem repeats. The greater similarities between the monomeric units and the Terminal Inverted Repeats (TIRs), Long Terminal Repeats (LTRs), and other non-coding regions of the transposable elements highlights the importance of these naturally repetitive segments for the emergence of new classes of repetitive DNA (Table 1; Figure 2).

The untranslated 5′ and 3′ (5′UTR and 3′UTR) extremities also serve as a source of satellite DNA. The evidence available for the pea, *Pisum sativum*, indicates that the satDNA PisTR-A originated through the amplification and homogenization of tandem repeats present in the hypervariable 3′UTR of the Ty3/gypsy-like Ogre elements [59]. The 3′ terminal may also play a key role in this process, as observed in *Drosophila melanogaster*, in which the emergence of new satDNA corresponded to the 3′ non-coding region of the transposable element HeT-A [87], and in the cetaceans, the DNA of the common satellite monomer is similar to the 3’-terminal portion of the mammalian L1 (LINE-1) retrotransposon [82] (Figure 2).

Open Reading Frames (ORFs) of TEs have also been found to be related to the satDNA in *Gallus gallus*, with similarities in the sequences being observed among the regions of the satDNA HinfI and the CR1 retrotransposon, which contains the partial ORF II of the CR1 element [7] (Figure 2). In the rodent *Phodopus roborovskii*, the ORF II of a LINE-1 retrotransposon is 88% similar to a small region of the PROsat present in the genome [81]. In rye, *Secale cereale*, the E900 satDNA contains a fragment of a retrotransposon which encodes a partial reading frame for the GAG-like protein of an LTR retrotransposon [75]. The GAG-like gene was also related to the CAA microsatellite in wheat, *Aegilops speltoides*, and presented similarities with the upstream region of the Ty3/gypsy-like retroelement [98] (Figure 2). Overall, then, the sum of the evidence indicates that any part of a transposable element may provide a substrate for the generation of new sequences, and there does not appear to be any conclusive evidence that specific portions of these elements are involved preferentially in the generation of satellites or other types of tandem repeat.

## 3. Is the Centromeric Region a Hotspot of the Emergence of de novo satDNA Derived from TEs?

Centromeric regions are favorable to the emergence and establishment of new families of satellite DNA and are ideal models for studying the TEs as an evolutionary substrate in chromosomal evolution [107,108]. Centromeres are gene-poor, and the majority of the transcriptional activity observed in this region involves non-coding RNAs that interact in the organization of the kinetochores [109].

Meiotic recombination is avoided in centromeric regions, which appears to be an evolutionary strategy to avoid chromosome aneuploidy, given that crossovers in the vicinity of the centromere make chromosomes more prone to mis-segregation [109,110]. The centromeres are thus a cold spot for crossovers, making this chromosomal region prone to the emergence of de novo satDNA from TEs. The mobile genetic elements inserted in these regions cannot be deleted easily by crossovers, and thus accumulate in the centromeric and pericentromeric regions [111].

Plant and animal centromeres are rich in TEs, and these sequences may sometimes be highly specific and/or involved directly in the architecture of this chromosomal region. Two species of beetle, *Dichotomius schiffleri* and *Tribolium castaneum*, have unusually extended centromeres, which appear to be related to the prevalence of DsGypsy6, LINE-1, and Helitron-like sequences in these structures [112,113]. In Poaceae species, the centromere is enriched with a specific retrotransposon family known as the centromeric retrotransposon (CR), which has highly conserved motifs, that are also found in their B chromosomes, when present (see [114]). The CR elements and centromere connection may contribute to the maintenance of the centromere/kinetochore complex, given that these sequences interact with the kinetochore protein CENH3 [115].

Recent studies of neo-centromeres have established a new perspective for the understanding of the role of TEs in the emergence of new satDNA families. Talbot and Hennikof [111] proposed a model which indicates an active role of the TEs in the evolutionary emergence of neo-centromeres. De novo centromeres appear to arise from the deposition of epigenetic markers in a particular region of the chromosome, followed by the enrichment of repetitive sequences at this new site [116]. The transposons appear to play an important role in the maintenance of the size of the region and the increase in the repeat content of the neo-centromeres that do not have many tandem repeats during this initial period [117].

Large numbers of CENP-A are detected flanking centromeres, but low levels of CENP-A may spread to the non-canonical centromere regions, predisposing them to acquire a potential centromeric function [118,119], and the insertion of TEs in an ectopic CENP-A site may create favorable conditions for the evolution of new a centromeric complex [118].

New cycles of transposition activity increase the density of TEs and may result in insertions of copies adjacent to the original, resulting in tandem duplications (tandem dimers, trimers, or more repetitions), as observed in the Rice8 satDNA in centromere of rice [120], for example. Retrotransposons comprise approximately 70% of the functional centromeres of *D. melanogaster*, and are composed of complex DNA rich in non-LTR retroelements inserted within large blocks of tandem repeats [121]. Some plant centromeres are composed of long arrays of satDNA interspersed with Gypsy LTR-retrotransposons [122,123,124,125], which highlights the interplay of the different classes of repetitive DNA in the establishment and maintenance of the centromere region.

Satellite DNA repeats derived from TEs in the centromere region have been identified in a number of different species [3,61,121,126]. The centromeres of *Zea mays* are a good example because it is composed of CRM1TR and CRM4TR satDNA, two tandem arrays that were derived from an LTR-retrotransposon in at least two separate events [96]. The repeat arrays of satellites in the centromeres of chromosomes 1, 2, 3, 5, 7, and 8 of the potato, *Solanum tuberosus,* were also amplified from retrotransposon sequences [95]. In the chicken karyotype, several centromere-specific types of satDNA are highly similar to retrotransposons [127]. In the centromeres of *Prunus* species, a highly conserved monomer unit of 166 bps has been identified from assembled genomes and sequencing reads, with varying signal intensities in fluorescence in situ hybridization (FISH) experiments, which indicates that the centromeric regions of this genus are enriched with this sequence [128].

The centromeric region of the Y chromosome of *Drosophila melanogaster* is composed of 18HT satDNA, a satellite DNA derived from HeT-A and TART non-LTR retrotransposons [87], the same elements that were co-opted to the telomeric function in *Drosophila*. In the *Drosophila obscura* species group (*Drosophila subobscura, D. guanche*, and *D. madeirensis)*, a DNA transposon (SGM elements) was the substrate for the emergence of a species-specific SGM satDNA that makes up the centromeric heterochromatin in *D. guanche*, in which the SGM element appears to be inactive [85].

The subtelomeric heterochromatin also appears to be prone to the emergence of tandem copies of transposable elements. *Drosophila melanogaster*, for example, has three telomere-specialized retrotransposons (HeT-A, TART, and TAHRE) that have been co-opted functionally to the organization of the chromatin and the maintenance of the telomere, and resemble telomere extensions containing telomerase [129,130]. The subtelomeric region of the *D. melanogaster* chromosomes presents a complex combination of potentially active elements and truncated TEs arranged in a long array [129]. In *Drosophila biarmipes*, by contrast, Helitron transposons may play an important role in the structure of the telomere, and while full-length Helitrons can be observed in the telomeres, fragments are interspersed within the abundant satellite sequences [129]. The richness of the repetitive arrays derived from the TEs in the subtelomeric heterochromatin of *Drosophila* represents a promising model for the study of the emergence of satellite DNA from TE elements and another potential chromosomal hotspot for the formation of de novo satDNA from TEs.

## 4. Mechanisms of the Production of Repeats from TEs

Unequal crossovers are an important mechanism of the expansion of tandem arrays and the homogenization of the satDNA by concerted evolution [131], although the absence of canonical meiotic crossovers in the centromeric region would require a mechanism that is independent of the emergence and evolution of the satDNA in this region. The transposition process may cause a Double Strand Break (DBS) of the DNA, which the DNA repair mechanism will attempt to fix, through alternative pathways, such as Non-Homologous End Joining (NHEJ), Non-Allelic Homologous Recombination (NAHR), or by Homologous Recombination (HR).

The NHEJ may contribute to the expansion of tandem arrays as a result of a single transposition event. Microhomology-mediated NHEJ has the potential to give rise to a wide range of chromosome events including duplications of sequences if it happens between sister chromatids [132] This mechanism also plays an important role in finalizing some TE-related instability events [133] and has been found with a dominant role in gene duplications [134]. TSD sequences are identical; therefore, it provides microhomology making it possible for NHEJ to restore the genome back to its original state before the insertion of a transportable element [135].

For example, Target Site Duplications (TSD) may be brought together by the NHEJ mechanism following a “cut and paste” transposition event (Figure 3), which rejoins the broken ends without the use of extensive homology [136,137]. As each TSD normally has fewer than 10 bps, these dimers of low complexity can arise easily through polymerase slippage and the subsequent expansion of the number of copies (Figure 3).

Studies on TSD and TE insertions were performed in D. melanogaster [1], where it was hypothesized that TEs organized in tandem could be generated through multiple insertions at the same chromosomal site, which also presents a TSD. The authors concluded, analyzing reads of the element known as P-element, that the junctions of the tandem repeats had the same size as the TSD of the said element, and the consensus motif was found similar to the TSD from other P-element insertions.

On the other hand, NAHR may also result in recurrent genome rearrangements, such as inversions, translocations, deletions, or duplications, in sequences that are not in intra- or inter-chromatid allelic positions, i.e., they are paralogous [138,139]. This mechanism provides valuable insights into the expansion of repeat arrays to the emergence of new satDNA, given that it depends on the position of the paralogous TE copies that will used as the template for the repair of the chromatid that has suffered the DBS, which may result in tandem duplications of the repetitive segments of the TE (Figure 3). Given this, intra- or inter-chromatid NAHR may contribute to the expansion of the initial repeat arrays in a manner similar to that of the unequal crossovers performed in the concerted evolution model.

One other especially interesting pathway of double-strand DNA repair is the use of a strand from a sister chromatid through Homologous Recombination (HR). Given either the existence of the repetitive elements that make up the molecular structure of the TEs (such as LTRs and ITRs) or the greater similarities of the TSD regions, the repair of DBSs using intrachromosomal HR mechanisms has the potential to generate tandem duplications in these regions [140] which may cause repeat arrays of the TE to accumulate in this region, if the Holliday junctions are processed to yield a crossover.

Similar results may also be obtained when a DBS is repaired by ectopic recombination, in which the paralogous copies used as the template are from non-homologous chromosomes [141]. In this case, the resolution of the Holliday junctions in the crossover events between dispersed repeats [142] will also result in the tandem duplications of the repetitive portions of the TEs in the region of the DBS. The outcome of these mechanisms of DBS repair is the formation of repeat arrays in the region of the DNA damage, which will be a source of other mechanisms of sequence expansion, such as polymerase slippage, that contribute to an increase in the number of monomers, leading to the emergence of satDNA from TE segments.

Finally, the intrinsic mechanisms involved in the transposition events are prone to the production of repeats derived from TEs. For example, the genesis of the Bari1 (Tc1-mariner superfamily) repeat clusters in *D. melanogaster* was provoked by anomalous rolling circle mechanisms and the subsequent reintegration within the *Stalker* LTR-retrotransposon, which generated an 80-repeat organized in heterochromatic clusters [143].

## 5. Conclusions

Repetitive DNA sequences have long been studied because of their structural role and impact on the genome of a range of different organisms. Here, however, we highlight the importance of also focusing on the relationships among the different elements. The possible presence of pre-existing transposable elements that serve as a substrate for the emergence of new families of repetitive sequences reinforces the potential importance of these elements in genome and karyotype remodeling during the evolutionary process. Given this, a better comprehension of the evolutionary relationships of these elements may be extremely valuable for a better understanding of the evolution of the eukaryote genome.

## Figures and Tables

**Figure 1 cells-11-03373-f001:**
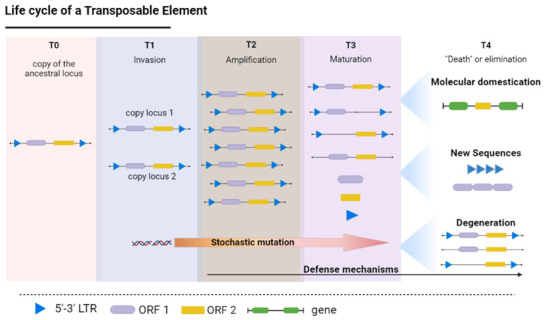
Life cycle of a transposable element, using LTR transposons with their respective Open Reading Frames (ORFs) as an example. **T0—**simple copy of the element in the genome; **T1—Invasion:** the chromosomal locus acquires at least one repeat unit; **T2—Amplification:** an increase in the number of copies present in the genome, either by burst events (in the case of class I elements) or the repair or homologous recombination of double-stranded DNA (class II); **T3—Maturation**: the elements are inactivated or silenced through epigenetic silencing, piRNAs, DNA methylation or other mechanisms, such as mutations; **T4—Death or degeneration**: the elements may be either eliminated from the genome, undergo molecular domestication, begin to exercise new functions or give rise to new repetitive sequences, such as tandem repeats. **LTR** = Long Terminal Repeat; **ORF** = Open Reading Frame. (Source: the authors / created in Biorender.com, accessed on 5 September 2022).

**Figure 2 cells-11-03373-f002:**
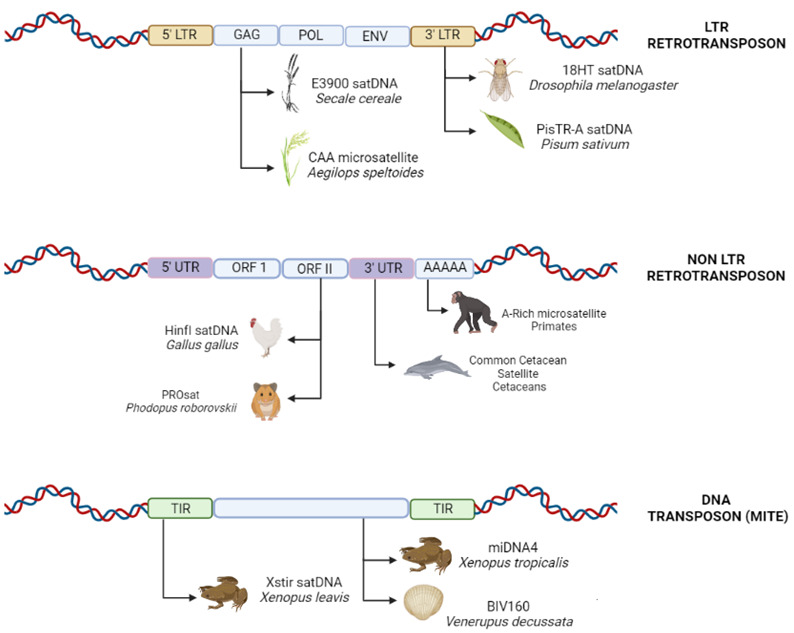
Schematic diagram of the different regions of transposable elements that may provide the starting point of origin for new micro-, mini- or satellite DNA. **LTR** (Long Terminal Repeat); **GAG** (GAG domain); **POL** (Reverse Transcriptase); **ENV** (Envelope Protein); **UTR** (Untranslated Region); **ORF** (Open Reading Frame); **TIR** (Terminal Inverted Repeats). (Source: the authors/created in Biorender.com, accessed on 5 September 2022).

**Figure 3 cells-11-03373-f003:**
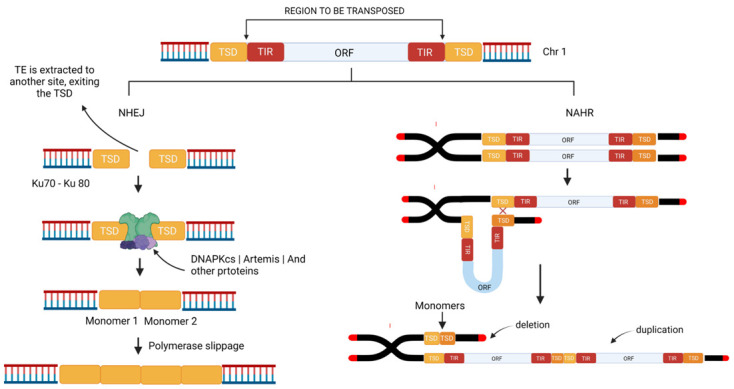
Schematic diagram showing how the Non-Homologous End-Joining (NHEJ) and Non-Allelic Homologous Recombination (NAHR) DNA repair mechanisms contribute to the expansion of tandem repeats from a Transposable Element. **TSD** (Target Site Duplication); **TIR** (Terminal Inverted Repeats); **ORF** (Open Reading Frame). (Source: Authors/created in Biorender.com, accessed on 5 September 2022).

**Table 1 cells-11-03373-t001:** Recorded cases of transposable elements giving rise to satellite, mini- and microsatellite DNA (satDNA), according to their mobility and the regions of the transposable elements that have given rise to the tandem repeat sequences. **ORF** (Open Reading Frame); **TR** (Tandem Repeat); **UTR** (Untranslated Region); **LTR** (Long Terminal Repeat), **TIR** (Terminal Inverted Repeats), **IR** (Inverted Repeat).

Species	Type of Transposable Element (TE)	Superfamily	Class	Mobility	Region of the TE	New Sequence	Reference
*Pan paniscus* and *Hylobates lar*	Alu elements	SINE Alu Family	Class I	non-autonomous	3′ oligo(dA) tail and A-rich middle region	A-rich primates’ microsatellites	[69]
*Homo sapiens*	Alu | LTR –retrotransposons	SINE Alu Family	Class I	non-autonomous	TR may occur in any region of the TE	7276 Minisatellites	[61]
*Homo sapiens*	Alu	SINE Alu Family	Class I	non-autonomous	-	pλg3, pMSl, pMS43, and pMS228	[66]
*Homo sapiens*	Alu	SINE Alu Family	Class I	non-autonomous	Near to 3′ -UTR.	Three minisatellites	[68]
*Homo sapiens*	Alu	SINE Alu Family	Class I	non-autonomous	3′ oligo(dA) tail.	(GAA)n	[67]
Mouse genome	SINE B1	SINE Superfamily	Class I	non-autonomous	GAGGCA dimmer within the SINE	(GGCAGA)n	[78]
Mouse genome	MaLR	Retrotransposon-like superfamily	Class I	non-autonomous	LTR	Ms6-hm e Hm-2	[79]
*Ctenomys* sp.	retroviral genome	-	Class I	-	LTR	RPCS satDNA	[80]
*Phodopus roborovskii*	LINE-1 elements	LINE	Class I	autonomous	ORF2	PROsat	[81]
Dolphin	LINE-1	LINE	Class I	autonomous	-3’ UTR	Common Cetacean Satellite	[82]
*Phoca vitulina concolour*	LINE-1	LINE	Class I	autonomous	ORF2	Pvc 20	[83]
Genus *Messor*	Mariner-like (*Mboumar)*	Tc1/mariner	Class II	autonomous	Mariner is found inside the satDNA	satDNA	[84]
*Gallus gallus*	CR1	CR1 family of LINEs	Class I	autonomous	-3′ UTR and a partial coding region of ORF 2	HinfI (SCR1)	[7]
*Helicoverpa zea*	HzSINE1 MITE-like	SINE Superfamily	Class II	non-autonomous	5′-IR	(GTCY)n	[72]
*Drosophila virilis, Drosophila americana,* and *Drosophila biarmipes.*	DINEs	Helitron	Class II	non-autonomous	Central tandem repeats (CTRs)	satDNA arrays	[8]
*Drosophila guanche*	SGM-IS	SGM Transposon Family	Class II	non-autonomous	-	SGM satDNA	[85]
*Drosophila virilis* group	pDv element	pDv transposable element family	Class II	-	Terminal repeat	pvB370 BamHI sat DNA	[86]
*Drosophila melanogaster*	TART | HeT-A	TART subfamilies of the HeT DNA family	Class I Class I	Autonomous non-autonomous	-3’ UTR	18HT satDNA	[87]
*Drosophila virilis*	Tetris	Foldback elements	Class II	non-autonomous	TIR	satDNA-arrays (TIR-220)	[88]
*Hydromantes imperialis* and *H. ambrosii*	SINE-like elements	SINE Superfamily	Class I	non-autonomous	tRNA-related region	Hy/Pol III	[89]
*Monopterus albus*	LTR- RetrotransposonGypsy	LTR-Retrotransposon-like	Class I	autonomous	LTR	satDNA MALREP	[90]
*Xenopus leavis*	Xmix MITE	-	Class II	non-autonomous	TIR	Xstir satDNA	[73]
*Xenopus leavis*	SINE-like	SINE Superfamily	Class I	non-autonomous	tRNA-related region	Satellite 1	[91]
*Xenopus tropicalis*	MITE of TC1-mariner	Tc1–Mariner	Class II	non-autonomous	stDNA located within the MITE element	miDNA4	[74]
*Ostrea edulis*	CvA	Pearl	Class II	non-autonomous	-	HindIII	[77]
*Venerupis decussata*	MITE (Pearl)	Pearl	Class II	non-autonomous	-	BIV160	[92]
*Arabidopsis thaliana*	Atenspm	En/Spm-like	Class II	autonomous	-5’ UTR	ENSAT1	[93]
*Pisum sativum*	Ty3/gypsy-like ogre	Gypsy	Class I	autonomous	-3′ UTR	PisTR-A satDNA	[59]
*Glycine max*	Gmr9/GmOgre	Ty3-gypsy	Class I	autonomous	Between the 3’UTR and Repetitive LTR	Gmr9-associate minisatellite	[94]
*Solanum bulbocastanum*	Sore 1	SORE-1 family	Class I	autonomous	LTR	Sobo satDNa	[6]
*Solanum tuberosus*	Ty3/gypsy-like	LTR-Retrotransposon-like	Class I	autonomous	LTR	St3-58; St3-238; St18; St3-294	[95]
*Zea mays*	CRM1 and CRM4	Centromeric Retrotransposons of Maize (CRM)	Class I	autonomous	UTR regions and LTR	CRM1TR e CRM4TR satDNA	[96]
*Hordeum vulgare*	BARE-1	BARE-1 retrotransposon family	Class I	non-autonomous	LTR	SSR	[97]
*Aegilops speltoides*	Cereba	Ty3-gypsy	Class I	autonomous	ORF to *gag*	CAA microsatellite	[98]
*Secale cereal* *e*	Crwydryn Tnr1 MITE	Crwydryn Tnr1/Stowaway family	Class I Class II	non-autonomous non-autonomous	ORF to *gag* *-*	E3900 satDNA D1100 satDNA	[75]
*Lathyrus sativus*	Ogre LTR-retrotransposons	LTR-Retrotransposon-like	Class I	autonomous	Close to the ORF to *gag*	nine satDNAs.	[99]
*Chenopodium album* aggregate	CACTA-Like *Jozin*	CACTA superfamily	Class II	non-autonomous	*Tnp2* TPase	CficCl-61-40 satDNA	[4]

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
