# Peer review of "Transposable Elements as a Source of Novel Repetitive DNA in the Eukaryote Genome"

_cells, 2022, doi:10.3390/cells11213373_

Round 1

Reviewer 1 Report

This is a review of the manuscript entitled “Transposable Elements as a source of novel repetitive DNA in eukaryotic genomes” submitted to Cells by Zattera and Bruschi. Overall, I found this review article scientifically stimulating and I recommend it for publication. I learned a great deal from this article, and I specifically appreciate how it uses examples from all eukaryotes, not just cancer cells or one organism. 

At the same time, this review article requires some work before it is ready to be published. I’ve separated my comments into major and minor changes to be made. 

Major changes to be made

1.     There is a significant amount of awkward writing. These are sentences where I can tell what the authors are attempting to say, but the language does not come out smoothly. Lines that should be rewritten for clarity include: 15-16 “…on the literature on TEs giving rise”; 18 “…arrays repeats as substrate to emergence of now repetitive elements in genomes”; 59-60 “…evolutionary lineages through split from ancestor species..”, etc...

2.     Related to point #1, the short paragraph on line 206 is only one sentence. Please either expand this into a paragraph to make the points, or add this point to another paragraph where it fits in. I prefer the expansion of this point into a full paragraph. 

3.     The paragraph starting on line 106 is about the role of DNA methylation in the repression of TE expression. However, histone modification is actually better conserved and functions to repress TE activity across eukaryotes. Please add sentences or a paragraph on histone modification to this review. 

4.     Section #3 starting on line 240 is very long. At the same time, this Review article is missing a section on “Mechanisms of TE-derived repeat production”. Please add such as section (with whatever title the authors see fit) to explore the molecular mechanisms responsible for making repeats from or with TEs. The important point here is “Mechanisms”. My suggestion is to break section #3 at line 304 and start the new section #4 on Mechanisms here. 

Minor changes to be made

1.     On line 74-76, a general statement about TE amplification is made, and then on line 77-78 a specific example is provided. However, the examples are from an odd organisms that I had to look up. Please explain to the reader if this idea is general to most eukaryotes, or specific to the few examples. 

2.     Reference 19 needs a title in the bibliography.

3.     As a note, I’ve never seen a “Methods” section for a literature review such as the one on line 154-160 before. This may not be needed, but is up to the authors. 

4.     Lines 310-311 HR should be “homologous recombination”

5.     Line 311 Target Site Duplication should be "TSD".

6.     Why is the term “Concerted Evolution” in capital letters? If this is a named proper noun for a theory / concept, please define this term in the review. 

Author Response

Response to Reviewer 1

[General Comment] This is a review of the manuscript entitled “Transposable Elements as a source of novel repetitive DNA in eukaryotic genomes” submitted to Cells by Zattera and Bruschi. Overall, I found this review article scientifically stimulating and I recommend it for publication. I learned a great deal from this article, and I specifically appreciate how it uses examples from all eukaryotes, not just cancer cells or one organism. At the same time, this review article requires some work before it is ready to be published. I’ve separated my comments into major and minor changes to be made. 

[Response] Thank you for your comments. We have gone through your concerns carefully and tried our best to address them one by one. We hope the manuscript has been improved accordingly.

[Major comment 1] There is a significant amount of awkward writing. These are sentences where I can tell what the authors are attempting to say, but the language does not come out smoothly. Lines that should be rewritten for clarity include: 15-16 “…on the literature on TEs giving rise”; 18 “…arrays repeats as substrate to emergence of now repetitive elements in genomes”; 59-60 “…evolutionary lineages through split from ancestor species..”, etc...

[Response] Thank you very much for pointing this out. We went through the entire manuscript, with the help of a proofreader, to eliminate grammatical mistakes and the awkward writing.

[Major comment 2] Related to point #1, the short paragraph on line 206 is only one sentence. Please either expand this into a paragraph to make the points, or add this point to another paragraph where it fits in. I prefer the expansion of this point into a full paragraph. 

[Response] Thank you very much for pointing this out. We expanded the sentence into a full paragraph. Please, see page 8 of the revised manuscript, lines 209-217. 

[Major comment 3] The paragraph starting on line 106 is about the role of DNA methylation in the repression of TE expression. However, histone modification is actually better conserved and functions to repress TE activity across eukaryotes. Please add sentences or a paragraph on histone modification to this review. 

[Response] Thank you very much for pointing this out. We wrote a sentence on histone modification. Please, see page 3 of the revised manuscript, lines 112-116.   

[Major comment 3]   Section #3 starting on line 240 is very long. At the same time, this Review article is missing a section on “Mechanisms of TE-derived repeat production”. Please add such as section (with whatever title the authors see fit) to explore the molecular mechanisms responsible for making repeats from or with TEs. The important point here is “Mechanisms”. My suggestion is to break section #3 at line 304 and start the new section #4 on Mechanisms here. 

[Response] Thank you very much for your suggestion. We add a new section on mechanisms, the title “Mechanisms of the production of repeats from TEs” can be seen on page 12 of the revised manuscript.

[Minor comment 1] On line 74-76, a general statement about TE amplification is made, and then on lines 77-78 a specific example is provided. However, the examples are from odd organisms that I had to look up. Please explain to the reader if this idea is general to most eukaryotes, or specific to the few examples. 

[Response] Thank you very much for pointing this out. We added a citation explaining the idea is general to most eukaryotes. Please, see page 12, lines 76-78 of the revised manuscript.

[Minor comment 2]   Reference 19 needs a title in the bibliography.

[Response] We apologize for our mistake. Revised accordingly.   

[Minor comment 3]   As a note, I’ve never seen a “Methods” section for a literature review such as the one on line 154-160 before. This may not be needed, but is up to the authors. 

[Response] We agree with the reviewer and chose to remove the “methods” section.

[Minor comment 4]   Lines 310-311 HR should be “homologous recombination”

[Response] We apologize for our mistake. Revised accordingly.  

[Minor comment 5] Line 311 Target Site Duplication should be "TSD".

[Response] We apologize for our mistake. Revised accordingly.  

[Minor comment 6] Why is the term “Concerted Evolution” in capital letters? If this is a named proper noun for a theory/concept, please define this term in the review. 

[Response] Thank you very much for pointing this out. Concerted evolution is a model explained on page 5, lines 195-199, but we agree there is no need for the capital letters, therefore, we removed it.

Reviewer 2 Report

In this review, the authors provide examples of satellite DNA and repeats originated from transposable elements in various organisms.

This is an interesting evolutionary topic that deserves a summary of the contemporary scientific literature.

The review is too scholastic in some of the passages and many of the papers cited have been reported in recent specific reviews.

In my opinion, the most relevant point of this review is paragraph 2.2 (There are portions of TE more prone to generate tandem repeat sequences? - needs grammar revision). However, this question remains unanswered at the end of the paragraph. Standing to the few scientific papers cited in this context, it seems that there are no preferred portions of TEs involved in the generation of satellites/repeats. I suggest to add one or few sentences to conclude paragraphs like this one, and let the readers have a clear message of what the authors think about.

The manuscript lacks citations to relevant scientific literature (please find some suggestions in the comments below).

Some concepts are almost cryptic, such as line 277-279. What is the connection between suppressed recombination rate and insertions adjacent to existing copies in the centromeric DNA?

There are grammar errors and typos throughout the text. I recommend revision of the manuscript with the aid of a proofreader with good written English skills.

In conclusion, I suggest to revise and improve the whole manuscript and try to make it more appealing to the readers. I will be glad to revise a modified version of the manuscript.

Below you can find additional specific comments.

L24 Not only tandem duplications contribute to the dissemination of paralogous copies, since paralogous can also be dispersed throughout the genome. I would suggest removing "tandem"

l50. "... Class II elements (DNA transposons) do not depend on such a molecule to carry out its mobility". I think this statement is wrong. RNA is always important for transposon activity, since 

it is at the basis of gene expression. Rather, I would say that Class II  elements do not relay on the retrotranscription mechanism. Similarly, I would remark that elements belonging to Class I reverse transcribe a RNA intermediate of the transposition.

L120-127. This paragraph discuss on the concept of heterochromatin as the cemetery of TEs. However, I would like to point out that there are several evidence for the role of TEs (both functional and molecular relics) in the heterochromatin, which are discussed in recent review (doi :10.3390/cells11050761)

l52-57. Although referred to Tc1/mariner elements, citation to the work by Hartl et al (doi: 10.1016/S0168-9525(97)01087-1) should be added at the end of this paragraph. Indeed, this model is assumed to be valid for all TEs.

l139. "....the evolutionary cycle life of TEs on the host genome..." not clear

l145 "evidence" is uncountable

l214. "Higher homology". since homology is not a quantitative concept, this should be changed to "higher similarity". If you still want to refer to homology just remove "higher".

l271. "..more CENP-A than needed is produced .." Can the authors justify this claim by citing a scientific work?

l217-276. this paragraph is not clear. Please revise.

l279 trimmers should be trimers

l307 "When a transposable element excises from one chromosomal site to another.." Excision does not require additional chromosomal loci. Please revise.

l314 “cut and past” should be “cut and paste”

There is a peculiar example of tandem repeat generation that is not mentioned in the text. 

The Bari1 repeat is a 80-copies repeat mapping in the heterochromatin of D. melanogaster. I think this repeat is unique since there is no other transposon repeat like this that has been described so far. The peculiar inter-monomer junction, the conservation of most of the transposon monomer (compared to the reference element), and the possible rolling circle replication-mediated origin (doi: 10.1023/A:1022916817285) make this transposon tandem repeat one of a kind. Furthermore this kind of repeat has been generated twice in D. melanogaster (doi: 10.1371/journal.pone.0156014), while it has not been detected in other Drosophila species, suggesting a possible host role in the generation of transposon-derived repeats.

The manuscript lacks a description of transposon-derived sub-telomeric and telomeric satellites, which is relevant to the review topic. Examples of telomeric satellites come not only from D. melanogaster (HET-A, TART, TAHRE) (do1:10.1101/gr.5348806), but also from other Drosophila species where different transposon types are possibly involved in the generation of telomeric satellites doi: (10.1101/gr.245001.118). Furthermore, many plant species display subtelomeric repeats made of transposable elements (doi: 10.1093/gbe/evw303; doi: 10.1073/pnas.0401243101).

Fgure 1 (bottom) "defende" should be "defense".

Figure 3. The left side of the figure shows a mode of repeat generation starting from a duplicated TSD, left by an excised TE. Although this option is theoretically feasible, can the authors discuss on previously published cases of repeats derived by expansion of TSDs?

The right side of the figure also have a problem. It is a reductive representation of the possible origin of tandem repeats. Moreover, there is no mention in the text of the frequency of occurrence of these events. 

Figure 3. the element depicted is designed as "transpososome". Is that correct?

Author Response

Response to Reviewer 2

[General Comment] In this review, the authors provide examples of satellite DNA and repeats originated from transposable elements in various organisms. This is an interesting evolutionary topic that deserves a summary of contemporary scientific literature. The review is too scholastic in some of the passages and many of the papers cited have been reported in recent specific reviews.

 [Response] Thank you very much for agreeing with us on the intention of this manuscript. We have read your comments carefully and tried our best to address them one by one. We hope that the manuscript has been improved toward Cells standards after this revision.

[Major comment 1]   In my opinion, the most relevant point of this review is paragraph 2.2 (There are portions of TE more prone to generate tandem repeat sequences? - needs grammar revision). However, this question remains unanswered at the end of the paragraph. Standing to the few scientific papers cited in this context, it seems that there are no preferred portions of TEs involved in the generation of satellites/repeats. I suggest to add one or few sentences to conclude paragraphs like this one, and let the readers have a clear message of what the authors think about.

 [Response] Thank you for your comment. We agree the question remains unanswered by the end of the section; therefore, we added sentences concluding the paragraph to let the message clearer. Please, see page 10 of the revised manuscript, lines 251 - 255.

[Major comment 2]   The manuscript lacks citations to relevant scientific literature (please find some suggestions in the comments below).

[Response] Thank you for the recommendation of scientific literature. They were added to the manuscript.

[Major comment 3]   Some concepts are almost cryptic, such as lines 277-279. What is the connection between suppressed recombination rate and insertions adjacent to existing copies in the centromeric DNA?

[Response]: We thank the reviewer for pointing this out. This sentence was written in the sense that, as the selection is expected to be weaker in regions with lower recombination, these regions are prone to TEs accumulation. But we agree this connection was not clear and we reformulated the paragraph. Please see page 11 of the revised manuscript, lines 295-303.

[Major comment 4]   There are grammar errors and typos throughout the text. I recommend revision of the manuscript with the aid of a proofreader with good written English skills.

 [Response]: Thank you for the nice reminder. We went through the entire manuscript, with the help of a proofreader, to eliminate grammatical mistakes.

 [Major comment 5]   In conclusion, I suggest to revise and improve the whole manuscript and try to make it more appealing to the readers. I will be glad to revise a modified version of the manuscript.

 [Response] We are grateful to the reviewer for your insightful comments on our paper, and we hope the manuscript after careful revisions, meets the expected standards. We look forward to hearing from you regarding our submission and to responding to any further questions and comments you may have.

[Minor comment 1]   L24 Not only tandem duplications contribute to the dissemination of paralogous copies, since paralogous can also be dispersed throughout the genome. I would suggest removing "tandem"

 [Response]:   Thank you for your reminder and correction. Revised accordingly.  Please see page 1 of the revised manuscript, line 24. 

[Minor comment 2]   l50. "... Class II elements (DNA transposons) do not depend on such a molecule to carry out its mobility". I think this statement is wrong. RNA is always important for transposon activity, since it is at the basis of gene expression. Rather, I would say that Class II  elements do not relay on the retrotranscription mechanism. Similarly, I would remark that elements belonging to Class I reverse transcribe a RNA intermediate of the transposition.

 [Response]: We agree with the reviewer’s assessment. We made the modifications for Class I and Class II regarding the RNA information. Please, see page 2 of the revised manuscript, lines 47-50.

[Minor comment 3]   L120-127. This paragraph discuss on the concept of heterochromatin as the cemetery of TEs. However, I would like to point out that there are several evidence for the role of TEs (both functional and molecular relics) in the heterochromatin, which are discussed in recent review (doi :10.3390/cells11050761)

[Response]: We appreciate the literature recommendation. We added this information to the manuscript. Please, see page 4 of the revised manuscript, lines 130-131.  

[Minor comment 4]   l52-57. Although referred to Tc1/mariner elements, citation to the work by Hartl et al (doi: 10.1016/S0168-9525(97)01087-1) should be added at the end of this paragraph. Indeed, this model is assumed to be valid for all TEs.

 [Response]: Thank you for recommending the work by Hartl et al. It was added at the end of the paragraph. Please see page 2 of the revised manuscript, lines 51-56.

[Minor comment 5]   l139. "... the evolutionary cycle life of TEs on the host genome..." not clear

 [Response] We thank the reviewer for pointing this out and we agree the sentence was not clear. Please see page 4 of the revised manuscript, lines 146-149.

[Minor comment 6]   l145 "evidence" is uncountable

[Response] We apologize for our mistake. Revised accordingly. Please see page 4 of the revised manuscript, line 151.

[Minor comment 7]   l214. "Higher homology". since homology is not a quantitative concept, this should be changed to "higher similarity". If you still want to refer to homology just remove "higher".

 [Response] We apologize for our mistake. Revised accordingly. Please see page 9 of the revised manuscript, lines 224-227.

[Minor comment 8]   l271. "..more CENP-A than needed is produced .." Can the authors justify this claim by citing a scientific work?

[Response] This information was provided by Leo et al (2020), where the authors claim that “it has been shown that CENP-A is produced in excess of the needs for centromeres and it is deposited around the original centromere and on islands of nucleosomes scattered along the chromosomes”. But we agree this paragraph was not clear and we reformulated it. Please see page 11 of the revised manuscript, lines 291-294.

[Minor comment 9]   l271-276. this paragraph is not clear. Please revise.

[Response] We thank the reviewer for pointing this out. Please see page 11 of the revised manuscript, lines 291-294, where the paragraph was reformulated it.

[Minor comment 10]   l279 trimmers should be trimers

[Response] We apologize for our mistake. Revised accordingly.

[Minor comment 11]   l307 "When a transposable element excises from one chromosomal site to another.." Excision does not require additional chromosomal loci. Please revise.

[Response] We apologize for our mistake. We rephrased the sentence to correct the error. Please, see page 12 of the revised manuscript, lines 345-348.

[Minor comment 12]   l314 “cut and past” should be “cut and paste”

[Response] We apologize for our mistake. Revised accordingly.

[Minor comment 13]   There is a peculiar example of tandem repeat generation that is not mentioned in the text. 

The Bari1 repeat is an 80-copies repeat mapping in the heterochromatin of D. melanogaster. I think this repeat is unique since there is no other transposon repeat like this that has been described so far. The peculiar inter-monomer junction, the conservation of most of the transposon monomer (compared to the reference element), and the possible rolling circle replication-mediated origin (doi: 10.1023/A:1022916817285) make this transposon tandem repeat one of a kind. Furthermore, this kind of repeat has been generated twice in D. melanogaster (doi: 10.1371/journal.pone.0156014), while it has not been detected in other Drosophila species, suggesting a possible host role in the generation of transposon-derived repeats.

 [Response]:  Thank you for recommending the literature and example. We have added the suggested content to the manuscript on page 14, lines 396-401.

[Minor comment 14]   The manuscript lacks a description of transposon-derived sub-telomeric and telomeric satellites, which is relevant to the review topic. Examples of telomeric satellites come not only from D. melanogaster (HET-A, TART, TAHRE) (do1:10.1101/gr.5348806), but also from other Drosophila species where different transposon types are possibly involved in the generation of telomeric satellites doi: (10.1101/gr.245001.118). Furthermore, many plant species display subtelomeric repeats made of transposable elements (doi: 10.1093/gbe/evw303; doi: 10.1073/pnas.0401243101).

[Response]:  Thank you for recommending the literature and examples. We have added the suggested content to the manuscript on page 11, lines 324-337.

[Minor comment 15]   Figure 1 (bottom) "defende" should be "defense".

[Response] We apologize for our mistake. Revised accordingly.

[Minor comment 16]   Figure 3. The left side of the figure shows a mode of repeat generation starting from a duplicated TSD, left by an excised TE. Although this option is theoretically feasible, can the authors discuss on previously published cases of repeats derived by expansion of TSDs?

[Response] We based this schematic representation using TSD on the work of McGurk and Barbash (2018), where the authors report a study about TSD and TE insertions in Drosophila melanogaster. This information was added to the text, as well. Please, see page 12 of the revised manuscript, lines 362-367.

[Minor comment 16]   The right side of the figure also have a problem. It is a reductive representation of the possible origin of tandem repeats. Moreover, there is no mention in the text of the frequency of occurrence of these events. 

[Response] Thank you very much for this comment. Although we know this is a simplified schematic model, we tried to elucidate more about this subject in the text. Please, see page 12, lines 350-356 of the revised manuscript.

[Minor comment 17]   Figure 3. The element depicted is designed as "transpososome". Is that correct?

[Response] Thank you for pointing this out. We changed the term “transposome” to “region to be transposed”.

Round 2

Reviewer 2 Report

I would like to thank the authors responding to all of my comments.

I think the manuscript has been improved in its readability and it now merits publication.

I would suggest to consider specifying, at lines 203-203, that you are referring to Class II elements. Although it could be obvious for TE folks, non-experts in the field would benefit from such information early in this sentence.

I would finally remove the term "segmental (line 599) since its evolutionary and structural significance is very different from the transposon tandem duplications that are described.

Author Response

Dear Reviewer,

We appreciate the time and effort that you and the reviewers dedicated to reviewing and providing valuable comments on our manuscript.

Please see below the point-by-point responses. All page numbers refer to the revised manuscript file with tracked changes.

Sincerely,

Daniel Pacheco Bruschi

[Reviewer] I would suggest to consider specifying, at lines 203-203, that you are referring to Class II elements. Although it could be obvious for TE folks, non-experts in the field would benefit from such information early in this sentence.

R= Added information.

[Reviewer] I would finally remove the term "segmental (line 599) since its evolutionary and structural significance is very different from the transposon tandem duplications that are described.

R= We agreed with the reviewer's suggestion and replaced the term "segmental duplications" by "tandem duplications" in all parts of the manuscript.